# A CNN-RNN Combined Structure for Real-World Violence Detection in Surveillance Cameras

**Soheil Vosta** 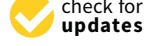 **and Kin-Choong Yow \***

Faculty of Engineering and Applied Science, University of Regina, Regina, SK S4S 0A2, Canada; sav323@uregina.ca
**\*** Correspondence: kin-choong.yow@uregina.ca

**Abstract:** Surveillance cameras have been increasingly used in many public and private spaces in recent years to increase the security of those areas. Although many companies still recruit someone to monitor the cameras, the person recruited is more likely to miss some abnormal events in the camera feeds due to human error. Therefore, monitoring surveillance cameras could be a waste of time and energy. On the other hand, many researchers worked on surveillance data and proposed several methods to detect abnormal events automatically. As a result, if any anomalous happens in front of the surveillance cameras, it can be detected immediately. Therefore, we introduced a model for detecting abnormal events in the surveillance camera feed. In this work, we designed a model by implementing a well-known convolutional neural network (ResNet50) for extracting essential features of each frame of our input stream followed by a particular schema of recurrent neural networks (ConvLSTM) for detecting abnormal events in our time-series dataset. Furthermore, in contrast with previous works, which mainly focused on hand-crafted datasets, our dataset took real-time surveillance camera feeds with different subjects and environments. In addition, we classify normal and abnormal events and show the method's ability to find the right category for each anomaly. Therefore, we categorized our data into three main and essential categories: the first groups mainly need firefighting service, while the second and third categories are about thefts and violent behaviour. We implemented the proposed method on the UCF-Crime dataset and achieved 81.71% in AUC, higher than other models like C3D on the same dataset. Our future work focuses on adding an attention layer to the existing model to detect more abnormal events.

**Keywords:** anomaly detection; surveillance cameras; ResNet; ConvLSTM; CNN+RNN; UCFcrimes

## 1. Introduction

With the many emerging challenges in public management, security, and safety, there is an increasing need for monitoring public scenes through surveillance cameras. At first sight, it seems an easy job for a human to monitor surveillance cameras feed to extract essential and helpful information from behavioral patterns, detect abnormal behaviours, and provide immediate response [1]. However, due to severe limitations in human ability, it is hard for a person to monitor simultaneous signals [2]. It is also a time-consuming task requiring many resources such as people and workspace [3]. Therefore, an automatic detection method is crucial to this end. One of the sub-domain in behaviour understanding [4] from surveillance cameras is detecting anomalous events. Anomaly detection in surveillance cameras is a challenging task that might face several problems: (1) abnormal events rarely happen; therefore, it is hard to find massive datasets of such events. This lack of samples might lead to some difficulties in the learning process. (2) Generally, everything that does not follow a specified pattern (or rule) is called an "anomaly". As a result, we cannot dedicate a model for abnormal events. (3) An action can be normal or abnormal in different situations. It means that even a global abnormal event (GAE) can be a routine activity in a particular situation like shooting in a gun club. The act of "shooting" is generally

considered abnormal, while it acts normal in a shooting club. On the other hand, some behaviour is not intrinsically abnormal, but it would be an anomaly in a specific location and condition called local abnormal event (LAE) [5]. Besides, Varadarajan in [6] proposed abnormal events as "an action done at an unusual location, at an unusual time".

From a learning standpoint, anomaly detection can be divided into three approaches: supervised, unsupervised, and semi-supervised, as a significant and well-known categorizing for learning methods. In supervised learning, there are two different approaches by considering whether the model is trained by only one category or all existing categories [7]. In other words, in single model learning, the model is trained by only normal(or abnormal) events, whereas in multi-model learning, both normal and abnormal events need to be trained. In the single model learning, anomalous events distinguished from normal ones by learning a threshold for normality definition [8–10], learning of a multidimensional model of normal events within the feature space [11–17] and learning rules for model definition [18]. While, for the multi-model learning approach, which is particularly used when there are several groups of anomalies, each class will be trained dependently or independently [7]. On the other hand, an anomaly detection problem is generally considered as an unsupervised learning problem [19]. This technique deal with unlabeled data in which it is assumed that Normal events frequently occur while Abnormal events rarely happen in data. Considering all rare events as anomalous is one of the drawbacks of this learning [7]. Several clustering algorithms in unsupervised learning consider normal and abnormal events should be well separated in the feature space [20–22]. Besides, the semi-supervised Anomaly detection approach neither is too reliable on labeled data like the supervised approach nor have a low accuracy as unsupervised models [23]. This model tries to diminish the differences between supervised and unsupervised techniques [19]. Several works take advantage of the properties of semi-supervised learning schema such as in [24–26]. This paper proposes the anomaly detection problem as a multiple scene formulation in a supervised learning schema. Numerous abnormal behaviours in the real world depend on our definition of an anomalous event to label it as an anomaly. However, here, we focus on the UCF-Crime dataset [27], including much abnormal, illegal, and violent behaviour captured by surveillance cameras in public places, which can lead to severe problems for individuals and a society population. Our proposed model used ResNet50 as a Convolutional Neural Network (CNN) for feature extraction. Then, due to working with the video dataset, we add an RNN, ConvLSTM, to the model architecture, which can work efficiently on such data to our model. Then, the model returns whether the input video includes illegal behaviour or not. This model can save humans time and money and increase the accuracy of detecting irreparable damages. Furthermore, the proposed model can significantly reduce the response time of emergency services, which is of paramount importance for governments and the population. The main contribution of the proposed method is summarized below.

- A combination of ResNet and ConvLSTM is used for anomaly detection from surveillance cameras.
- We used the UCF-Crime dataset, which includes natural scenes recorded by surveillance cameras in 13 categories of abnormal events.
- To better understand each anomaly category, we defined two modified datasets from UCF-Crime by splitting the normal scenes from the abnormal ones.

In the rest of the paper, we analyze the other related works which use different models and each sub-models of the whole idea for anomaly detection in surveillance cameras in Section 2. Then, in Section 3, we describe our proposed model. Next, evaluate our work by several experiments in Section 4. Finally, the conclusion and future works are discussed in Section 5.

## 2. Related Works

With the widespread use of surveillance cameras and the emerging need for automatically abnormal event detection, several methods have been proposed to solve various types

of anomaly detection in video datasets [5–9]. Supervised learning methods aim to separate data classes, whereas unsupervised techniques explain and understand data characteristics. Between the two, supervised anomaly detection techniques outperform unsupervised anomaly detection techniques using labeled data [28]. In supervised anomaly detection, the separating boundary is learnt from training data, and then test data are classified into either normal or abnormal classes using the learned model.

In 2015, Tran et al. [29] presented a model for learning spatiotemporal features with 3D convolutional networks. This model is called C3D, and in this model, each segmented video goes through a three convolution layer 3D ConvNet to classify different actions. After four years, Sultani et al. [30] used this model with multiple instance learning (MIL) in their paper to find abnormal events.

However, deep neural network architecture has recently been successfully used in various computer vision tasks, including anomaly detection problems. Mainly, supervised deep learning for anomaly detection includes two parts: a feature extraction network followed by a classifier network [31]. This paper implements Convolutional Neural Networks (CNN) to extract essential features from each input video data frame. By taking advantage of the Recurrent Neural Network (RNN) structure, the system can investigate a series of frames to find any abnormal events.

### 2.1. Convolutional Neural Networks

CNNs are the most popular choice of neural network for the image processing goals [32]. Extracting complex hidden features from high dimensional data with a complex structure is the main advantage of CNNs, making them suitable feature extractors for sequential and image datasets [33,34]. The extracted deep features were utilized in different applications like image quality assessment [35], skin lesions classification [36], and person re-identification [37]. Although CNNs are widely used in various deep learning tasks like text classification and NLP, they are mainly used in computer vision, such as for image and video detection and classification [38]. Various kinds of CNNs have been built in the recent decade like AlexNet, ResNets, VGG, Inceptions and their variants. Several works were also done by combining these convolutional neural networks with a softmax layer [39], and morphological analysis [40] in the anomaly detection area. In addition to CNN, Xu et al. [41] and Hasan et al. [42] proposed autoencoder structures. Nguyen et al. [43] proposed a Bayesian nonparametric approach for abnormal event detection in video streams. Moreover, several other models like Fisher vector and PCA [44], Motion Interaction Field (MIF) [45] have been proposed in this scope.

However, there is also some model that is mainly designed for focusing on more than one dimension of data.

One of the most common CNN used for feature extraction in deep learning methods is ResNets. A regular CNN is typically a combination of convolutional and fully connected layers [46]. The number of layers depends on several criteria, and each kind of CNNs has its structure. For instance, AlexNet has eight layers, and GoogleNet has 22 layers. Another type of Artificial Neural Network called Residual Neural Network (ResNet) has a somehow different structure. ResNet uses skip connection (or shortcuts), which can jump over layers. The main reason for using such shortcuts is to pass activations from previous layers to subsequent layers for better memorizing the parameters, which leads to diminishing the chance of vanishing gradients [47].

### 2.2. Recurrent Neural Networks

On the other hand, RNNs is one of the well-known choices for capturing features in analyzing time series data [48]. However, they fail in extracting context as time steps increases. Long short-term memory (LSTM) networks were introduced to overcome this limitation by improving the long-term dependency in RNNs [49]. Due to the sequential nature of the surveillance camera feeds, LSTM networks have become more popular for anomaly detection applications [26]. Therefore, several researchers worked on anomaly

detection problems using the LSTM structure. Using regularity scores from reconstruction errors in an LSTM-based network is one approach of using LSTM to solve anomaly detection problems [50,51]. Furthermore, Srivastava et al. proposed a model using autoencoders, the encoder LSTM, and Decoder LSTM in an unsupervised learning approach [52].

However, the only RNNs methods could not achieve high accuracy results. They mainly predict the subsequent frames in a video time-series and, by calculating the difference between the ground truth and predicted value, decide whether the video segment is abnormal or not. Therefore, as the abnormal events do not follow a particular algorithm, it is difficult to say an uncommon event happened based on the prediction of the next frame.

### 2.3. CNN + RNN

Deep learning architectures perform well in learning spatial (via CNNs) and temporal (via LSTMs) features individually. Spatiotemporal networks (STNs) are networks in which spatial and temporal relation features are learned [53]. In STNs, both CNNs and LSTMs are combined to extract spatiotemporal features [31]. After applying CNN to the data, the output of the CNN structure (ResNet or AlexNet, for instance) will be the input of the subsequent LSTM. Several researchers adopt such techniques for detection in video dataset like [30,54,55] for finding abnormal events. Furthermore, another approach has emerged in recent years in which a convolutional layer filters the output of CNN before entering the LSTM structure [50,56,57]. This new approach is called Convolutional LSTM or ConvLSTM. As a result, instead of fully connected in LSTM, a convolutional layer dramatically decreases the number of parameters. Therefore, the chance of overfitting decreases, and it can boost the model's performance.

### 3. Proposed Method

Now, we will build our proposed model by using the mentioned concepts. We implement Residual Networks (ResNets) as one of the most efficient techniques for feature extraction in deep neural networks [47]. Then, in the next phase, we use Convolutional LSTM (ConvLSTM) as a recurrent network (RNN) to find the anomalies in our video dataset. The whole idea, as shown in Figure 1, indicates that each video file is divided into sequences of $n$ frames, and the difference between each frame and the very next one is the input to the CNN (i.e., ResNet50). The output of the ResNet50 will then go to the RNN (i.e., ConvLSTM). After this process is done for all $n$ frames, the output goes to a max-pooling layer followed by several fully connected layers to achieve the outcome.

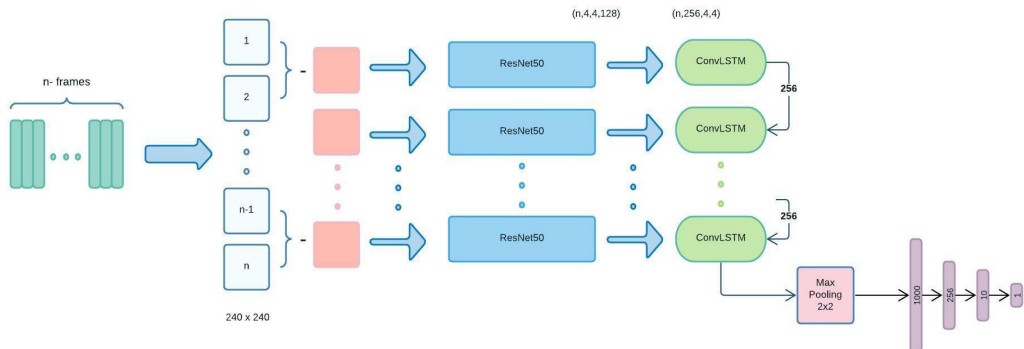

**Figure 1.** ResNet50ConvLSTM structure.

Now, we go through the proposed model in detail in the following.

### 3.1. Preprocessing

In the first step, we divide each video file into fixed frames. In that, if we are going to divide a video file into $n$ frames out of the total number of the video file's frames, we calculate the number of frames that should be skipped. Therefore, if the length of a video file is 60 s, the total number of video frames will be $m = 1800$ if the video format is set to

30 f/s. Now, assume that $n = 30$ we need to select 30 frames out of 1800 frames. Thus, we need to select each frame after 60 skipped frames. After selecting the frames, we calculated the difference between each frame and its adjacent one to consider the spatial movement for each input. We also preprocessed one group of datasets in another way. We selected three categories of the UCF-Crime dataset. We split the exact time of the abnormal event for each video file and labeled them as "Anomaly", so the rest of each video was labelled as "Normal". Then, divide each video file into the same length file (e.g., 5 s). Consequently, as in the previous scenario, $n$ frames will be selected from $m$ frames, but in this case, the number of $m$ is also fixed like $n$. Therefore, in the contract of the previous work, we focused only on the frames that abnormal events are happening. Besides, the normal set is also taken from the same file that includes the anomalous event. It means that, in contrast of the original UCF-Crime dataset, the background, lighting, and objects are all the same, and only the acting is different, which helps the system detect abnormal events better.

### 3.2. ResNet50

ResNets showed an outstanding performance on several well-known datasets, e.g., ImageNet [58], and is known as one of the most common models in many applications in various fields of machine learning, such as action recognition. Although, there are several types of ResNets with different layers, like ResNet-18,26,50,101,152. We chose ResNet50 in our proposed model because of its easy to understand the structure and better performance, among other methods, considering its complexity. Due to difficulties in collecting and labeling anomalous events, we used Transfer Learning in our model [59]. Therefore, we pre-train the model on the ImageNet dataset, including 1000 image categories. Therefore, by running ResNet50 on ImageNet, the parameters will be initialized and updated, and the model is ready to run on our desired dataset.

In our case, as shown in Figure 1, the difference of each frame and the very next frame of each video file goes to a ResNet50. The original input image frame size is $240 \times 240$. Thus, the input of our ResNet50 will be ($240 \times 240 \times 3$) with "channel_last" data format. Then, after passing through several convolutional and pooling layers, the output for Deep Residual Features (DRF) is a 4d tensor (n, 1,1,2048) and need to be reshaped for the ConvLSTM filters. Figure 2 illustrates the structure of ResNet50 that is used in our model. So, the ResNet50 output reshapes to (n, 4,4,128) and is ready to pass the ConvLSTM layer. As we do not need to classify with ResNet, we do not use a 1000-d fully connected layer. We only use the output of the last convolution block for DRF extraction to work on an extracted feature in the next layer [60].

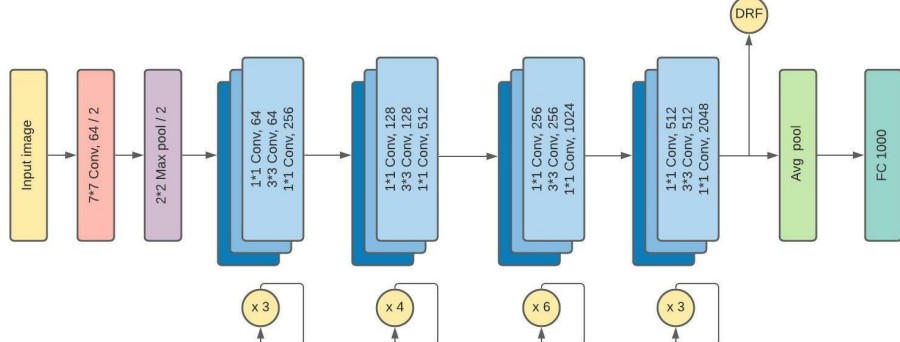

**Figure 2.** ResNet50 structure.

### 3.3. ConvLSTM

As the input, cell output, and states of LSTM or, more particularly, Fully-Connected LSTM (FC-LSTM) are all 1-D vectors, they cannot preserve the spatial relationship between pixels in images and videos. Therefore, LSTM is not suitable for spatial sequence data [57,61]. The very first application of ConvLSTM was in forecast prediction using radar images [57]. However, in recent years, it has been applied to other applications which

are mainly focused on prediction in computer vision tasks like slip detection [62]. In the ConvLSTM approach, because of the convolutional layers, all the inputs, cell outputs, states, and the spatial dimensions in the last two dimensions (rows and columns), are 3-D tensors. Therefore, as ConvLSTM has convolutional gates in its structure, it can provide the model with spatial and temporal changes. The better performance of ConvLSTM in comparison with regular LSTM was proved by several experiments in [57]. Figure 3 shows ConvLSTM structure in details. Where $X$ is the inputs and $C$ is cell output. Hidden states shown by $h$, and $i_t, f_t, o_t$ are the gates in the ConvLSTM structure. As a result, it is clear that for this paper's goal, to find abnormal events dependent on spatial and temporal features, the best option is to use ConvLSTM to gain a more efficient result.

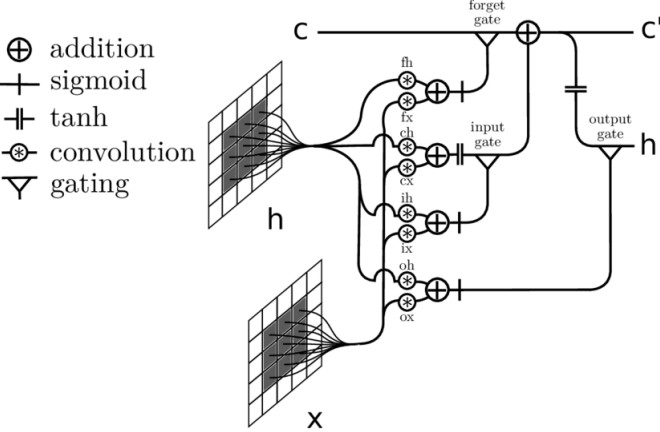

**Figure 3.** ConvLSTM structure [63].

From a complexity standpoint, as LSTM works with 1-d vectors and so after the Hadamard product, it makes too many parameters that increase the complexity of the model and enhance the chance of occurring overfitting. As a result, due to the advantages of ConvLSTM unique structure, it needs fewer parameters, reducing the model's complexity and preserving the spatial relationships, which are reasonably fit for images and videos.

Each frame after passing from ResNet enters to a ConvLSTM cell including 256 hidden state (filters) with the kernel size $= (3 \times 3)$. The input for our ConvLSTM is a 4D tensor, (n, filter size = 256, row = 4, column = 4) so that the input of each time step is a $4 \times 4$ image with 256 channels.

### 3.4. Classification

Finally, in the fourth layer, the output of the last ConvLSTM goes to a maxpooling layer with size $(2 \times 2)$. Then, the result will be flattened to achieve a 1-d vector. Next, the output vector passes through several fully-connected layers with batch normalization and ReLU activation. Consequently, if we are only willing to do binary classification, after going through 1000-d, 256-d, and 10-d fully-connected layers, we use sigmoid activation and binary_crossentropy for the loss function, and then figure out whether the final output shows an Abnormal event or not. However, if our final classification goal is more than two categories, we use softmax activation with categorical_crossentropy as a loss function.

### 3.5. ResNet+ConvLSTM

So far, we have discussed ResNet and ConvLSTM, where each works well in the tasks they are designed for. In this section, we propose our method by adding the ConvLSTM model to the ResNet structure, in that the output of ResNet goes through to the ConvLSTM network.

The following describes the whole idea and procedure of the proposed ResNet50ConvLSTM model shown in Figure 1.

- At first, each input video is divided into $n$ frames. Then, the difference between each frame and the next frame goes to preprocessing phase as an initial part of the ResNet structure, including data augmentation, de-noising, and normalizing.
- Next, the difference of each two frames goes to our ResNet50 structure. It passes through 34-layers of Batch-Normalization, Convolutional, pooling and fully-connected layers to provide a suitable feature extraction of a vector size of 1000.
- Subsequently, the output for each image will be the input for the related ConvLSTM. Each ConvLSTM layer will be provided with two inputs: one from the previous ConvLSTM (256 feature maps) and the second from the data processed by ResNet50. As a result, this structure is significantly helpful for spatial-sequential data, particularly videos. The ConvLSTM that we used in our model consists of 256 filters with the size of $3 \times 3$ and stride 1.
- Finally, the output of the last ConvLSTM layer, which includes all information from previous stages with size $(n, 4, 4, 128)$, goes to a $(2 \times 2)$ MaxPooling layer followed by some fully connected layers to provide the desired classification.

## 4. Experimental Results

In this section we compare our experimental results with other methods applied on the UCF-Crime to evaluate how good our model works. We used AUC and Accuracy metrics for our evaluation.

### 4.1. Dataset

In this paper, we implement the proposed model on the UCF-Crime dataset [27] including lots of abnormal, illegal and violent behaviour captured by surveillance cameras in public places like schools, stores, and streets. The reason for selecting this dataset is that this dataset is extracted from actual everyday events that can happen every day and everywhere. Moreover, these kinds of abnormal behaviours can lead to severe problems for individuals and society. Several papers use some handcrafted dataset or a particular dataset with the same background and environment (i.e., hockey fight dataset and movie dataset), which is rarely found in our daily routine. This dataset includes long untrimmed surveillance camera feeds in 13 categories of anomalous events (i.e., Abuse, Arrest, Arson, Assault, Road Accident, Burglary, Explosion, Fighting, Robbery, Shooting, Stealing, Shoplifting, and Vandalism) as well as Normal events category. Figure 4 indicates part of the UCF-Crime dataset. To have a fair comparison with other works in this field, we used 75% of data for training and 25% for testing in our experiments.

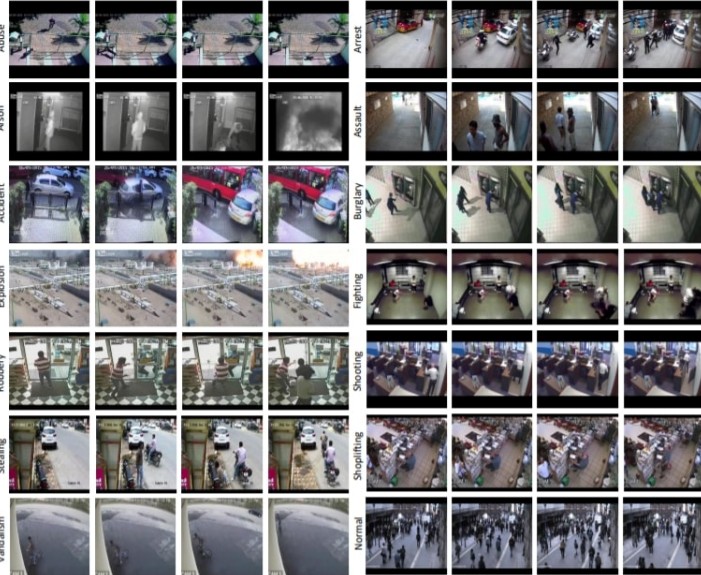

**Figure 4.** Examples of UCF-Crime dataset.

We used four variants of the UCF-Crime dataset: Ucfcrimes, Binary, 4MajorCat, and NREF. The Ucfcrimes is our original dataset in which we have 14 classes. While in the Binary, we consider all 13 abnormal events as one class of anomalous. In the 4MajorCat, we categorized the abnormal events into three big groups versus one normal group. Therefore, the three groups named Theft (i.e., Burglary, Robbery, Stealing, Shoplifting), Vandalism (i.e., Arson, Road Accident, Explosion, Vandalism), and Violence behaviours (i.e., Abuse, Arrest, Assault, Fighting, Shooting). The NREF is another manipulated data that focuses only on three anomalies (Road Accident, Explosion, and Fighting). In this dataset, instead of using predefined abnormal and normal videos, in each abnormal video, the segments include anomalies considered abnormal. In contrast, the rest of the video file is labeled as normal. Besides, we trimmed all videos in NREF into 10 seconds videos to have more related frames from each file. The number of videos for each category for Ucfcrimes and Binary datasets are shown in Table 1. Furthermore, Table 2 indicates the number of videos for 4MajCat and NREF datasets.

**Table 1.** Number of videos for Binary and Ucfcrimes datasets.

| Binary | No. Videos | Ucfcrimes | No. Videos |
|---|---|---|---|
| Abuse | 50 | Abuse | 50 |
| Arrest | 50 | Arrest | 50 |
| Arson | 50 | Arson | 50 |
| Assault | 50 | Assault | 50 |
| Burglary | 100 | Burglary | 50 |
| Explosion | 50 | Explosion | 50 |
| Fighting | 50 | Fighting | 50 |
| RoadAccident | 150 | RoadAccident | 50 |
| Robbery | 150 | Robbery | 50 |
| Shooting | 50 | Shooting | 50 |
| Shoplifting | 50 | Shoplifting | 50 |
| Stealing | 100 | Stealing | 50 |
| Vandalism | 50 | Vandalism | 50 |
| Normal | 950 | Normal | 50 |
| Total | 1900 | Total | 700 |

**Table 2.** Number of videos for 4MajCat and NREF datasets.

| 4MajCat | No. Videos | NREF | No. Videos |
|---|---|---|---|
| Theft (Burglary, Robbery, Shoplifting, Stealing) | 150 | RoadAccident | 30 |
| Vandalism (Arson, Explosion, RoadAccident, Vandalism) | 150 | Explosion | 50 |
| Violence behaviours (Abuse, Arrest,Assault, Fighting, Shooting) | 150 | Fighting | 70 |
| Normal | 150 | Normal | 150 |
| Total | 600 | Total | 300 |

### 4.2. Experimental Settings

Our experiments applied the proposed model using ResNet50 plus ConvLSTM, which are included in Keras library. We utilized several hyperparameters for tuning the model to achieve the best result of our experiments. Table 3 compares the results of our experiments with different values for using data augmentation or not as well as the type of initial weight and optimizer. Consequently, we applied data augmentation on our dataset, and utilized glorot_uniform for initial weights and RMSprop as an optimizer in our experiments.

**Table 3.** Tuning hyper parameters for binary classification on the UCF-Crime dataset.

| Hyper Parameters | Tune | Acc (%) |
|---|---|---|
| Data Augmentation | True | 54.18 |
| Data Augmentation | False | 53.82 |
| Initial Weight | glorot_uniform | 63.88 |
| Initial Weight | random_uniform | 54.17 |
| Initial Weight | he_uniform | 54.17 |
| Optimizer | RMSprop | 63.88 |
| Optimizer | Adam | 61.34 |

Furthermore, we used glorot_uniform for initial weights and RMSprop as an optimizer in our experiments. Moreover, the learning rate of our model is set to $10^{-4}$. The epochs are also set to 50, but the code stops once the loss function convergences and does not make more sense when it continues. It means, after each epoch, it calculates loss function, so if the difference of loss functions of two subsequent epochs is less than the tolerance value, the process stops, and the last accuracy will be the result. We also need to define the number of frames we take from each video file. We set this sequence length to 20 in our experiments. Regardless of its size, each video file is divided into 20 frames by considering the same interval of skipped frames.

### 4.3. Evaluation

In our experiments, we used different kinds of evaluations. In the first step, We tested our model with several types of CNN models (VGG19, InceptionV3, ResNet50, ResNet 101, and ResNet 152) in keras library. Table 4 shows the comparison. Therefore, we selected ResNet50 for our experiments due to good accuracy with the less complicated structure.

**Table 4.** Comparison among different type of CNN in keras for binary classification.

| Model | Acc (%) |
|---|---|
| ResNet50ConvLSTM | 62.5 |
| InceptionV3ConvLSTM | 62.5 |
| VGG19ConvLSTM | 59.32 |
| ResNet101ConvLSTM | 63.75 |
| ResNet152ConvLSTM | 56.25 |

We compare our proposed method with a 3D convolutional network by measuring values; Accuracy (Acc) and Area under the curve (AUC). As the UCF-Crime dataset is relatively a new dataset, there are not too much work on that yet. Table 5 compares the AUC value for binary classification on the UCF-Crime dataset for our proposed method (ResNet50ConvLSTM) and seven other models for anomaly detection such as, SVM, MIL [30], C3D [29], and TSN [64]. We consider all mentioned abnormal event categories only as one category, 'Anomaly', and other data with no abnormal events as 'Normal'. The test classifier shows the probability of the correct classification for abnormal events. Therefore, as shown in Table 5, our model outperformed the previous methods.

Figure 5 depicts the training curve for binary classification with regards to accuracy and loss value.

In case of accuracy, we also compared our model with C3D by considering all 14 categories (13 abnormal events plus one normal event). In order to illustrates how the proposed model worked on the UCF-Crime dataset, we evaluate the method by calculating *precision*, *recall*, and *F1-score* that can be seen in Table 6. Furthermore, the provided confusion matrix in Figure 6 shows the classification details.

**Table 5.** AUC for binary classification on the UCF-Crime dataset.

| Model | AUC(%) |
|---|---|
| SVM Baseline | 50.0 |
| Hasan et al. [42] | 50.6 |
| Lu et al. [65] | 65.51 |
| Sultani et al. (loss without constraints) [30] | 74.44 |
| Sultani et al. (loss with constraints) [30] | 75.41 |
| Zhong et al. (C3D) [64] | 81.08 |
| Zhong et al. ($TSN^{Optical flow}$) [64] | 78.08 |
| our proposed model | 81.71 |

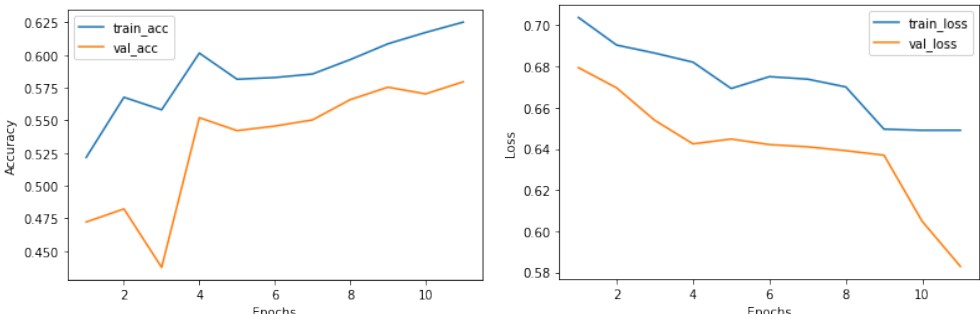

**Figure 5.** Training curve for binary classification.

**Table 6.** Evaluation for 14 categories classification on the UCF-Crime dataset.

| Evaluation Metric | (Value%) |
|---|---|
| *Precision* | 22.93 |
| *Recall* | 24.31 |
| *F1-score* | 23.60 |
| *Accuracy* | 22.72 |

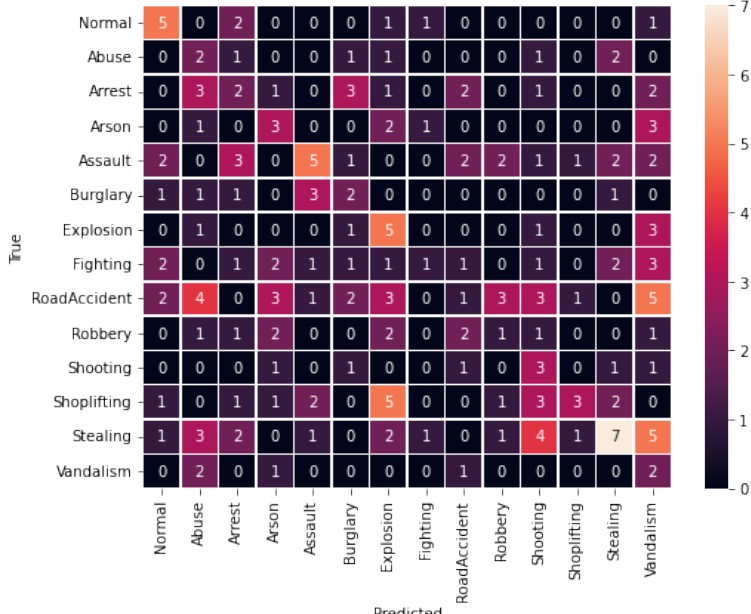

**Figure 6.** Confusion matrix for 14 categories classification on the UCF-Crime dataset.

In that case, as the Ucfcrimes have many everyday actions in real situations with different lighting and angles, it is difficult to distinguish them with an outstanding accuracy value. The proposed method works relatively as good as C3D. However, this sophisticated

dataset structure might need a more complicated design for feature extraction. It can be happened by using more convolutional layers to access more high-level features. In this case, we used ResNet101 instead of ResNet50 in our proposed model, which increased the classification accuracy. Therefore, by using ResNet101, our model, we achieved a fractionally better accuracy over the C3D model in our experiments. Table 7 shows this comparison.

**Table 7.** Accuracy for 14 categories classification on the UCF-Crime dataset.

| Model | Acc (%) |
|---|---|
| ResNet50ConvLSTM | 22.72 |
| ResNet101ConvLSTM | 23.75 |
| C3D [29] | 23 |

As we cannot achieve an outstanding accuracy in classifying all types of anomalies in the Ucfcrimes dataset, we categorized them into four major groups: Theft, Vandalism, Violence, and Normal behaviours.

The results for calculating Accuracy and AUC for these four major categories are shown in Table 8. Therefore, as we can see, we got a pretty much better result for the 4MajCat dataset, approximately 40 percent on accuracy compared to the accuracy for classifying all categories. Table 8 also indicates the result for NREF(Normal, Road accidents, Explosions, and Fighting). This experiment shows that with considering this video trimming, the improvement of classification increased significantly.

**Table 8.** Accuracy and AUC for all four datasets variants.

| Dataset | AUC (%) | Acc (%) |
|---|---|---|
| NREF | 79.04 | 65.38 |
| 4MajCat | 73.88 | 62.22 |
| Ucfcrimes | 53.88 | 22.72 |
| Binary | 81.71 | 62.5 |

## 5. Conclusions, Limitations and Future Work

This paper defines a novel structure combining ResNet50 and ConvLSTM to detect abnormal behaviour in the UCF-Crime dataset. There are several limitations that we faced in implementing this model. The dataset we used is in different illumination, speed, and subjects. For instance, some anomalies happened in the video, while we could not see any person in some videos (i.e., car accidents). Besides, we need to deal with another limitation of our dataset. The abnormal events may only take one or two seconds to happen, and even in 10-second videos, more than 80 per cent of the video length shows it is normal behaviour. Despite all mentioned limitations, our proposed method outperforms other methods on the UCF-Crime dataset. In addition to using all 14 categories of UCF-Crime, binary classification, and dividing into four major categories, we also trimmed the original video of three different anomalous events. We have both abnormal and normal events with the same background and objects. We implemented one of the most popular CNNs, ResNet50, to extract the most critical features from each input video frame. Then, the output of each ResNet goes through a ConvLSTM structure to explore the abnormal event in a series of frames. Finally, we used classifiers for each dataset to figure out how the model accurately recognizes each input video's right category. Although the experimental results show that our approach did better than other exiting models, we are looking to improve classifying all 13 types of anomalies in the UCF-Crime dataset. One of the approaches is adding an attention layer to the structure that we will work on in the future. Therefore, this attention layer can be added to CNN structure and/or ConvLSTM. Thus, the model can focus more accurately on the happening anomalies in the video file.

**Author Contributions:** Conceptualization, S.V. and K.Y.; methodology, S.V. and K.Y.; software, S.V.; validation, S.V.; formal analysis, S.V.; investigation, S.V.; resources, S.V.; data curation, S.V.; writing—original draft preparation, S.V.; writing—review and editing, S.V.; visualization, S.V.; supervision, K.Y.; project administration, K.Y.; funding acquisition, K.Y. All authors have read and agreed to the published version of the manuscript.

**Funding:** We acknowledge the support of the Natural Sciences and Engineering Research Council of Canada (NSERC), funding reference number DDG-2020-00034. Cette recherche a été financée par le Conseil de recherches en sciences naturelles et en génie du Canada (CRSNG), numéro de référence DDG-2020-00034.

**Data Availability Statement:** The UCF-Crimes dataset includes 1900 long untrimmed videos captured by surveillance cameras in which, half of them contain real world abnormal events while the rest are normal videos. It covers 13 kinds of anomalies including Abuse, Arrest, Arson, Assault, Road Accident, Burglary, Explosion, Fighting, Robbery, Shooting, Stealing, Shoplifting, and Vandalism. The data presented in this study are available in [30].

**Conflicts of Interest:** The authors declare no conflict of interest.

### Definitions

The following definitions of the evaluation metrics are used in this manuscript:

- AUC: The AUC (Area under the curve) of the ROC (Receiver operating characteristic; default) or PR (Precision Recall) curves are quality measures of binary classifiers. This class approximates AUCs using a Riemann sum. During the metric accumulation phrase, predictions are accumulated within predefined buckets by value. The AUC is then computed by interpolating per-bucket averages. These buckets define the evaluated operational points.
- *Accuracy*: Calculates how often predictions equal labels. This metric creates two local variables, total and count that are used to compute the frequency with which $y\_pred$ matches $y\_true$. This frequency is ultimately returned as binary accuracy: an idempotent operation that simply divides total by count.
- *Precision*: Calculated by $\frac{TP}{(TP+FP)}$ where $TP$ means the number of True Positives, $FP$ the number of False Positives.
- *Recall*: Calculated by $\frac{TP}{(TP+FN)}$ where $TP$ means the number of True Positives, $FP$ the number of False Negative.
- *F1-score*: The *F*1 score is the harmonic mean of the precision and recall. In that, as it close to 1 it shows the better value while 0 means the worst value for *F1-score*. This metric calculated from *precision* and *recall* value as follow:

$$F1 = \frac{2 \times (precision \times recall)}{(precision + recall)}$$

- Confusion Matrix: Provides a summary of predicted results on a classification problem, and shows them in a matrix format to depicts the number of correct and incorrect predictions.

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
