# Peer review of "A CNN-RNN Combined Structure for Real-World Violence Detection in Surveillance Cameras"

_applsci, doi:10.3390/app12031021_

Round 1
Reviewer 1 Report
The manuscript "A CNN-RNN combined structure for real-world violence detection in surveillance cameras" presents a spatio-temporal model based on the combination of features extracted using CNN and RNN networks. The authors conducted experiments on four variants of the UCF-crime dataset and claim that the proposed model outperforms the well-known C3D architecture.
Overall, the manuscript presents solid ideas with a vast number of experiments. Nevertheless, the manuscript requires major corrections in order to be considered for publication.
A main concern about the work is how it is written. There are several grammatical mistakes, and the readability of the text could be improved by adding the corresponding connectors between the sentences.
For example, in the sentence "Although many companies still recruit someone to monitor the cameras, but it is totally wasting time and energy" the connector "but" is misapplied. Additionally, it is unclear why hiring someone for monitoring is not desirable.
Another example is:
"In recent years surveillance cameras have been increasingly used in many public and private spaces. They mainly use to increase the security of that area. Although many companies still recruit someone to monitor the camera"
In this example, the sentences are not connected, which makes it difficult to understand. Also, there are grammatical errors, including a missing comma after the word "years" and missing auxiliaries ("They mainly use" should be "they are mainly used").
The authors also abuse of quantifiers like "many," which add ambiguity to the manuscript.
There are also omitted references, such as "several methods have been proposed to solve various 64 types of anomaly detection in video datasets."
Finally, the UCF-crime dataset is now a benchmark dataset, so I consider that the authors should compare their work with other state-of-the-art methods and not only the C3D architecture.
Reviewer 2 Report
In this manuscript, the authors designed a model by using well-known convolutional neural network (ResNet50) for extracting important features of each frame and a recurrent neural network for detecting abnormal events in time-series datasets. The manuscript is generally well-written and it is easy to follow.
I see the following weak points of the submitted manuscript and I have the following concerns:
1.) From line 76 to 82, the authors write about the potential of deep features extracted from CNNs. However, they omit to mention some examples. The authors should mention here examples. For example, deep features were utilized for image quality assessment (Multi-pooled Inception features for no-reference image quality assessment, AppSci, 2020), skin lesions classification (Deep features to classify skin lesions), or person re-identification (Deep-person: Learning discriminative deep features for person re-identification, 2020).
2.) In the Introduction section, a separate Contributions subsection would be helpful for ascertaining the main contribution of the manuscript.
3.) The authors write: "We implement Residual Networks (ResNets) as one of the most efficient techniques for feature extraction in deep neural networks" Could the authors explain why ResNet are the most efficient? What does "efficient" mean in this context? Why for example Inception-V3 is not so effective?
4.) The structure of the proposed ConvLSTM is not so clear. Although Figure 3 depicts a detail of ConvLSTM but the number of layers, the number of neurons in each layer is not clear from Figure 4. Probably, one more figure would make clear the structure.
5.) The authors write: "Also, we used glorot_uniform for initial weights and RMSprop as an optimizer in our experiments." Could other weight initialization or optimizer improve the performance?
6.) The definition of the evaluation metrics should be given in the manuscript.
7.) Since the applied dataset in Table 4 has more categories, the publication of a confusion matrix could be helpful for the readers.
8.) Since deep learning involves a lot of experiments, the curves of the training process should be also given.
Reviewer 3 Report
This is the review report of the paper entitled "A CNN-RNN combined structure for real world violence detection in surveillance cameras".
The paper presents a very important topic and is well presented. However, I have some comments to improve the paper.
1- The used dataset is small, the model could be overfitted. I would suggest using any new dataset to prove the model hasn't been overfitted and well generalized.
2- I would suggest reporting the highest achieved results in the abstract.
3- The section "2.1. Convolutional Neural Networks" required more recent papers. I would suggest the following one
https://link.springer.com/article/10.1186/s40537-021-00444-8
4- Paper code with a nice demo is important to upload on any public platform.
5- The contributions of the article have to be clear for the readers, I would suggest making them as bullet points at the end of the introduction.
6- Comparison with previous methods ( recent ones) is needed on the same used dataset of previously published papers with suitable references.
7- visualize what the ResNet learned at least for the first and last layers.
8- Have you used Transfer Learning from ImageNet when ResNet has been used? elaborate more on that.
8- The results are promising and can be improved. I would suggest using same-domain transfer learning for the lack of training data issue. I would suggest the following paper for the same-domain TL
https://www.mdpi.com/2076-3417/10/13/4523
9- I support your future work by adding an attention layer which I believe will help.
10- Accuracy is not enough to evaluate, list other measurements such as recall, precision, F1-score
Round 2
Reviewer 1 Report
Authors responded appropriately to the suggestions offered to improve the manuscript's overall quality. Ia ppreciate the time they spent to get this new version on time.
In the first version of the manuscript, I had two significant concerns about it. On one hand, the readability of the manuscript does not allow a clear understanding of what the authors did and why it was important. Now, in this revised version, the authors provide a well-written text that eliminates the majority of the remarks. Nevertheless, there are still minor points to be solved. For example,
- The abstract does not appear to have gotten the same level of attention as the manuscript's content; it reads more like a collection of sentences rather than a coherent sequence.
- There are a few typographical errors in the text. Occasionally, the authors used "UCF-Crimes" rather than "UCF-Crime". Similarly, like in line 94 (",skin lesions.."), some spaces are omitted. In line 304, a "the" is missing between "on" and "UCF-Crime".
- The headers in Table 1 are not displayed correctly.
On the other hand, although the authors conducted a set of valuable experiments, there was still a lack of a proper comparison with the state-of-the-art approaches. In this improved version, the authors compare the proposed model with seven models, finding that they outperform the selected works. However, the author does not mention why they chose these papers for comparison and how this selection validates the contribution of their approach.
Based on the above comments, I consider that the manuscripts still require minor corrections in order to be considered for publication.
Reviewer 2 Report
The authors addressed very carefully my comments and concerns. I recommend this manuscript for publication.
Author Response
Thank you very much for your consideration and your comments.
Reviewer 3 Report
The authors have addressed the comments very well,
I have one one concern, values in Table 5 are pretty low which made confused, is that good enough to publish or there is explantion for that.
